# Measurement Invariance across Sexual Orientation for Measures of Sexual Attitudes

**DOI:** 10.3390/ijerph20031820

**Published:** 2023-01-19

**Authors:** Laura Elvira Muñoz-García, Carmen Gómez-Berrocal, Alejandro Guillén-Riquelme, Juan Carlos Sierra

**Affiliations:** 1Mind, Brain, and Behavior Research Center (CIMCYC), University of Granada, 18011 Granada, Spain; 2Faculty of Health Sciences, Valencian International University (VIU), 46002 Valencia, Spain

**Keywords:** sexual attitude, sexual orientation, measurement invariance

## Abstract

Despite the growing interest in the study of sexual attitudes across sexual orientation, few studies have tested whether the instruments used to measure them are invariant. This study examined measurement invariance (configural, weak, strong, and strict) across sexual orientation in three different sexual attitude scales: the Sexual Opinion Survey (SOS) to assess erotophilia, the Hurlbert Index of Sexual Fantasy (HISF) to assess attitudes toward sexual fantasies, and the Negative Attitudes Toward Masturbation Inventory (NATMI) to assess negative attitudes toward masturbation. A total of 2293 Spanish adult men and women with different sexual orientations (i.e., heterosexual, bisexual, and gay) participated in the study. The results indicated strict invariance for HISF across sexual orientation and only weak invariance for SOS and NATMI. Differential item functioning was also found in two items of the NATMI scale. Evidence of validity was provided for the three scales that were studied.

## 1. Introduction

In general, an attitude is a psychological state that precedes and directs behavior [1]. Sexual attitudes are evaluative beliefs that determine favorable or unfavorable responses to sexual stimuli and influence both sexual functioning and the way of living or expressing sexuality [2,3]. Depending on the object or topic being addressed, sexual attitudes can be general or specific. In this sense, while erotophilia implies an attitude toward sexuality in general, specific sexual attitudes refer to behaviors or specific psychological processes related to sexuality (e.g., attitudes toward sexual fantasies or masturbation). Both erotophilia and attitudes toward specific sexual behaviors are associated with sexual health indicators such as sexual victimization [4,5,6], sexual aggression [7,8], sexual risk behaviors [9,10], the subjective experience of orgasm [11,12], and sexual functioning [3,13,14]. This relationship pattern indicates that sex-positive attitudes are associated with greater desire toward a sexual partner [15,16,17]; greater subjective and objective sexual arousal [18,19]; and greater facility in obtaining orgasm [3], along with greater intensity in its subjective experience [11,20] and greater satisfaction with it [21] as well as greater sexual satisfaction in general [15,22].

The erotophobia–erotophilia construct is defined, according to Fisher et al. [23], as the learned disposition to respond to sexual stimuli on a bipolar dimension of affect and evaluation from negative (erotophobia) to positive (erotophilia) poles. Based on this conceptualization, the Sexual Opinion Survey (SOS) [23] makes it possible to classify people as erotophilic or erotophobic. Erotophilic individuals place their responses in the scores closer to the positive pole of the evaluative continuum, i.e., they value sex as pleasurable and seek sexual activity, whereas the responses of erotophobic individuals are placed in scores closer to the negative pole, i.e., they value sexual activity negatively and tend to avoid it [23]. Erotophilia is considered an indicator of sexual health [18,24,25], as it is related to overall good sexual functioning [13,26,27,28,29] and, specifically, with sexual desire [16,30,31,32], subjective and objective sexual arousal [18,19,33,34], the propensity for sexual arousal [35,36], the subjective experience of orgasm [11], and sexual satisfaction [15,22,37]. Based on the Sexual Opinion Survey (SOS) [23], Vallejo-Medina et al. [28] developed a brief Spanish version composed of six items grouped into a single factor called erotophilia. In this scale, higher scores indicate greater erotophilia. The SOS was selected because of its adequate psychometric properties, as it is a measure that is invariant by gender, age, having a partner or not, and level of education. Moreover, its internal consistency reliability is above 0.75, and it has presented adequate evidence of validity [18].

Sexual fantasies refer to any mental image that is sexually arousing or erotic for the individual [38]. A positive attitude toward sexual fantasies is considered an indicator of sexual health [39], as it is directly related to sexual functioning [3,27], the subjective experience of orgasm [12], sexual satisfaction [40], sexual assertiveness [41] and positive attitudes toward masturbation [42]. The Hurlbert Index of Sexual Fantasy (HISF) [43] was developed to assess attitudes toward sexual fantasies. The scale was adapted to Spanish by Desvarieux et al. [44] and consists of a single factor where high scores show a favorable attitude toward sexual fantasies. This instrument was used because of its adequate psychometric properties. It has been shown to be invariant by gender, age, and educational level, with an internal consistency reliability of 0.94. It has also shown the ability to differentiate between individuals with and without difficulties in sexual functioning and has shown expected correlations with related constructs (erotophilia, sexual assertiveness, sexual desire, and sexual functioning) [3,14].

Specific sexual attitudes that consider masturbation as an object of evaluation have also been considered. Masturbation has traditionally been stigmatized [42,45]; however, it is a pleasurable sexual experience, and in recent decades there is evidence that masturbation is an indicator of sexual development and a means to achieve sexual health [46]. In this sense, a favorable attitude toward masturbation has been related to erotophilia and positive attitudes toward sexual fantasies [47], while a negative attitude is directly associated with sexual guilt [48,49], lower solitary sexual desire, lower satisfaction with orgasm [21], and worse sexual functioning [42]. One of the existing scales to measure attitudes toward masturbation is the Negative Attitudes Toward Masturbation Inventory (NATMI) [50,51], which Cervilla et al. [42] adapted to the Spanish population, proposing a brief one-factor version with high reliability (ordinal alpha of 0.95). This measure was used because of its adequate psychometric properties since this version allows discrimination between people with different levels of sexual desire and arousal, facility for reaching orgasm, and satisfaction associated with orgasm [42]. Higher scores on this scale indicate negative attitudes toward masturbation.

Studies on the sexual attitudes of LGB people are scarce when compared to similar research on heterosexual people. Overall, the body of collected evidence indicates differences in sexual attitudes across sexual orientation. More specifically, bisexual people are more erotophilic than heterosexual people [52,53] and show greater sexual openness [54]. Likewise, in comparison with heterosexual people, non-heterosexual people present a more positive attitudes toward sexuality [52], and bisexual and lesbian women report more permissive sexual attitudes and intentions than heterosexual women [32,55,56]. From these results, we can derive the importance of measuring invariance across sexual orientation for measures of sexual attitudes that are frequently used.

Measurement invariance testing is used to evaluate the assumption that scales operate similarly across groups [57,58], which constitutes one of the current guidelines of the International Test Commission [58]. Given that the description and explanation of how the dimensions that conform sexuality work involves an analysis through groups and collectives defined, among other criteria, by gender and sexual orientation, measurement invariance testing acquires an unquestionable importance in the field of human sexuality. In this sense, the production in recent years reflects the interest in examining the factorial invariance of scales that assess different dimensions of sexuality [12,18,25,42,59,60,61,62,63,64,65,66,67]. The purpose of this study was to examine the measurement invariance across sexual orientation of the Spanish versions of three scales that assess sexual attitudes: the Sexual Opinion Survey (SOS) [23] by Vallejo-Medina et al. [28], the Hurlbert Index of Sexual Fantasy [43] by Desvarieux et al. [44], and the Negative Attitudes Toward Masturbation Inventory (NATMI) [51] by Cervilla et al. [42].

## 2. Materials and Methods

### 2.1. Participants

The sample consisted of 2293 Spanish adults (1093 men and 1200 women) aged 18 to 77 years (*M* = 35.34; *SD* = 13.36). The participants were classified according to their scores on the Kinsey Scale into heterosexual people (scores = 1 and 2; *n* = 800), bisexual people (scores = 3, 4, and 5; *n* = 694), and gay people (scores = 6 and 7; *n* = 799). The inclusion criteria included being cisgender (Table 1).

### 2.2. Instruments

The Sociodemographic and Sexual History Questionnaire collects information on gender, age, nationality, education level, partner relationships, sexual activity, and masturbation.

The Kinsey Scale [68] identifies the type of sexual relationship practiced through an item with seven response options ranging from 1 (*exclusively heterosexual*) to 7 (*exclusively homosexual*).

The reduced Spanish version of the Sexual Opinion Survey (SOS) [23] by Vallejo-Medina et al. [28] measures erotophilia and consists of six items (e.g., “I personally find that thinking about engaging in sexual intercourse is arousing” [“*Me resulta excitante pensar en tener una relación sexual coital*”]) that are answered on a seven-point Likert scale from 1 (*strongly disagree*) to 7 (*strongly agree*). Higher scores indicate a higher level of erotophilia. In this study, the internal consistency reliability coefficient was 0.64. 

The Spanish version of the Hurlbert Index of Sexual Fantasy (HISF) [43] by Desverieux et al. [44] consists of ten items (e.g., “I think sexual fantasies are healthy” [“*Considero saludables las fantasías sexuales”*]) with a five-point Likert response scale from 0 (*never*) to 4 (*all of the time*). A higher score indicates a more favorable attitude toward sexual fantasies. In this study, the internal consistency reliability coefficient was 0.87.

The Spanish version of the Negative Attitudes Toward Masturbation Inventory (NATMI) [51] by Cervilla et al. [42] consists of ten items (e.g., “I feel guilty about masturbating” [“*Cuando me masturbo me siento culpable”*]) with a five-point Likert response scale from 1 (*not at all true for me*) to 5 (*extremely true for me*). High scores indicate negative attitudes toward masturbation. In this study, the internal consistency reliability coefficient was 0.82.

### 2.3. Procedure

Participants answered the survey online, which is a common procedure to assess sexual attitudes [3,42]. The online survey was distributed using virtual platforms (Facebook, Twitter, WhatsApp, and e-mail). Previous studies have confirmed that there are no differences compared to the traditional paper and pencil method [69,70].

Participants accepted an informed consent form that included the objective and purpose of the study. Anonymity, data protection, and confidentiality were guaranteed. Automatic responses were avoided by answering a random arithmetic question. The study was conducted according to the guidelines of the Declaration of Helsinki and approved by the University of Granada Human Research Ethics Committee (2594/CEIH/2022).

### 2.4. Data Analysis

Missing data were imputed through an algorithm for non-parametric distributions by creating a random forest model for each variable. Overall confirmatory factor analyses (CFAs) and an evaluation of measurement invariance via multiple groups (heterosexual, bisexual, and gay) were performed. CFAs were conducted in R^®^ (version 3.6.3) [71] with the Rstudio^®^ (version 1.2.5042) [72] interface. The following packages were used: missForest (version 1.4) [73] for missing data imputation and lavaan for the measurement of invariance [74]. The robust weighted least-squares estimation method with a chi-square fit with respect to the mean (WLSM) was followed. A fit was considered good if the root-mean-squared error of approximation (RMSEA) values were less than 0.06 and the comparative fit index (CFI) and Tucker–Lewis index (TLI) values were greater than 0.90. The factorial invariance was progressively analyzed at four levels: configural, weak, strong, and strict. To accept the equivalence of the models at the different levels, the recommendations on the CFI as the main invariance adjustment were followed. A change in CFI equal to or greater than 0.01 allows adopting the less constrained model and rejecting the more restrictive one [75,76].

For the differential item functioning (DIF) analysis, the logistic regression procedure was used, applying three models (fitting the overall score, including the item, and total item interaction) in the group prediction. The models were progressively subtracted to determine if there were significant increases in the explained variance (Nagelkerke R^2^ ≥ 0.035). For items with DIF, the response categories were progressively grouped to analyze whether the DIF was constant across all response options. The lordif package [77] was used for the analysis.

## 3. Results

### 3.1. Item Analysis

An analysis of the medians, mean scores, and the standard deviations of the items was conducted to detect extreme scores [78,79].

Regarding the SOS, means ranged between 5.52 and 6.59 (theoretical range of 1 to 7). Standard deviations ranged from 0.92 to 1.96. This indicates that the subjects had a response pattern with high frequencies at the higher (erotophilic) end of the scale (Table 2). 

In regard of the HISF, means ranged between 2.66 and 3.42 (theoretical range of 0 to 4). Standard deviations ranged from 0.72 to 1.19. This indicates that the subjects had a response pattern with high frequencies in the middle range of the scale (Table 2). 

With respect to the NATMI, means ranged between 1.02 and 1.30 (theoretical range of 1 to 5). Standard deviations ranged from 0.14 to 0.75. This indicates that the subjects had a response pattern with high frequencies at the lower (positive attitudes toward masturbation) end of the scale (Table 2).

### 3.2. Overall Confirmatory Analysis

The results of the overall CFAs are reported in Table 3. Initial CFAs in the full sample indicated an acceptable model fit for the hypothesized factor unique to SOS, HISF, and NATMI.

### 3.3. Measurement Invariance across Sexual Orientation

The factorial structure of the SOS did not show strict invariance by sexual orientation. It stayed at the weak level (RMSEA = 0.045 [0.036, 0.054], CFI = 0.936) (Table 4). 

The factorial structure of the HISF showed strict invariance by sexual orientation (RMSEA = 0.092 [0.088, 0.096], CFI = 0.904) (Table 4). 

The factorial structure of the NATMI did not show strict invariance by sexual orientation. It stayed at the weak level (RMSEA = 0.021 [0.017, 0.026], CFI = 0.947) (Table 4). 

### 3.4. Differential Item Functioning (DIF)

The NATMI scale presented uniform DIF in items 5 and 8. The values obtained in Mellenberg ranged between 0.029 and 0.080 for item 5 and between 0.038 and 0.080 for item 8. Therefore, in both cases the DIF occurred equally in all response categories. The differences in DIF according to trait level can be seen in Figure 1. The rest of the NATMI items, as well as the SOS and HISF items, did not present DIF problems (Table 5).

## 4. Discussion

Erotophilia and attitudes toward specific sexual behaviors (e.g., sexual fantasies and masturbation) are associated with indicators of sexual health [4,7,9,12] and with components of sexual functioning [16,18,22]. The few studies conducted with LGB population indicate that their sexual attitudes may differ from those reported by heterosexual individuals. The main objective of this study was to examine the measurement invariance by sexual orientation of the Spanish versions of three sexual attitude scales: the Sexual Opinion Survey (SOS) [28], the Hurlbert Index of Sexual Fantasy (HISF) [44], and the Negative Attitudes Toward Masturbation Inventory (NATMI) [42]. 

The results of the measurement invariance analysis of the HISF confirmed that it is a scale that is invariant by sexual orientation. That is, the measure of positive attitude toward sexual fantasies provided by this instrument can be compared across sample groups defined according to their sexual orientation (e.g., heterosexual, bisexual, and gay) [80]. This finding is consistent with the results of previous studies that indicated that the content of sexual fantasies is very similar among heterosexual, bisexual, and gay individuals [38,81,82,83,84]. This is especially useful in the treatment of sexual dysfunctions since the acceptance of sexual fantasies and support for opening up about them with a partner are important tools in sex therapy, enhancing personal growth and intimacy in the relationship [85]. Moreover, the results of this study indicate that both SOS and NATMI only reached a weak level of invariance. Reaching a strong level of invariance implies that the mean differences in the construct capture all mean differences in the shared variance of the items. When the overall model fit is significantly worse in the strong invariance model compared to the weak invariance model, as occurred in these two scales, it is understood that at least one item differs in the compared groups [76]. Therefore, a DIF analysis of both scales was performed. In the case of the SOS scale, no differential functioning of any of its items was observed. The absence of invariance in this scale may be due to the presence of extreme mean scores [86] for heterosexual, bisexual, and gay people. Specifically, a distinct tendency toward erotophilia was observed for each of these groups.

In the case of the NATMI scale, the DIF analysis reported that two items (“I feel guilty about masturbating” [“*Cuando me masturbo me siento culpable*”] and “When I masturbate, I am disgusted with myself” [“*Cuando me masturbo me doy asco*”]) did not perform as well as the rest, which implies a significant difference between heterosexual, bisexual, and gay people in the information they report about the feeling of guilt associated with solitary masturbation. Historically, guilt has been found to be associated with this behavior [45,87] and preceded by a negative attitude toward masturbation [48]. This guilt has been rarely compared between gay and heterosexual people, finding worse attitudes toward masturbation and greater associated guilt in heterosexual people than in gay people [88]. In the case of this scale, extreme mean scores were also found in the responses of the groups. In heterosexual, bisexual, and gay people, a clear tendency toward a positive attitude toward masturbation was observed, which may have caused the absence of measurement invariance [86]. The presence of extreme mean scores could reveal that the three groups established by sexual orientation (i.e., heterosexual, bisexual, and gay) share a generalized cultural consent [89] on attitudes toward aspects of sexuality in general and toward masturbation in particular. However, this is not the case for the HISF scale, where a more central and less extreme distribution of the means on the response scale was observed.

Beyond the psychometric explanation, which offers a logical justification for the non-invariance of the SOS and NATMI scales, the results obtained in this study must have a background based on the nature of the functioning of attitudes [90]. Specifically, in the HISF scale, each of its items refers to the abstract term “sexual fantasies” (e.g., “I think sexual fantasies are healthy” [“*Considero saludables las fantasías sexuales”*]). These sexual thoughts can be experienced with a positive or negative affect [39,91]. The experience that a person has had with respect to an attitudinal object, such as sexual fantasies in this case, determines the attitudinal affective component linked to that object [92]. Therefore, the abstract concept “sexual fantasy” to which the items of the HISF scale allude, depending on the socialization that the person has had with respect to these sexual fantasies or thoughts, will determine their positioning on the response scale, and this positioning will surely follow a normal distribution, providing sufficient variability. However, unlike the HISF scale, the SOS and NATMI scales present items that refer to very specific sexual concepts and practices with clear positive or negative associated evaluative components (e.g., “Masturbation can be an exciting experience” [*“La masturbación puede ser una experiencia excitante”*], and “I feel guilty about masturbating” [“*Cuando me masturbo me siento culpable”*], respectively).

In conclusion, this study provides evidence of the validity of the measures provided by the SOS, HISF, and NATMI scales, despite the absence of strong invariance by sexual orientation in SOS and NATMI. Considering the validity indicators of these scales, both reached the levels of configural and weak invariance, so they maintain their factorial structures and constitute accurate assessment instruments.

The limitations of this study include the fact that the sample was collected incidentally through an online survey that was distributed through social media, which excludes the participation of people without access to social media. However, beyond these limitations, the obtained results are considered relevant from a research and clinical perspective, as they provide evidence of validity regarding instruments of relevance in both settings. In the future, when creating and validating assessment instruments, a diverse sample with respect to sexual orientation should be considered. Further research should also consider the possibility of reproducing the analyses carried out in this study with other existing scales related to sexual attitudes, such as the revised Sociosexual Orientation Inventory (SOI-R) [93].

## 5. Conclusions

The results of this study support the strict invariance by sexual orientation of the HISF scale, which measures attitudes toward sexual fantasies, thus providing evidence of validity that allows comparisons of evaluations of this construct among heterosexual, bisexual, and gay people. Evidence of the validity and reliability of the SOS and NATMI scales is also provided, although strict invariance by sexual orientation was not found. Therefore, in the case of using them to make comparisons between sexual orientation groups, extreme caution should be applied.

The results of this study provide highly relevant information to the field of study of sexual attitudes in bisexual and gay people, showing that attitudes regarding erotophilia, sexual fantasies, and masturbation are rather similar to those of heterosexual people. On a more practical level, this study provides evidence of validity for the SOS, HISF, and NATMI questionnaires to be used in a clinical setting on LGB people. Analyzing these attitudes that are closely related to sexual health provides a more detailed understanding of the causes of certain LGB sexuality-related problems and acts as a guide to their potential solutions.

## Figures and Tables

**Figure 1 ijerph-20-01820-f001:**
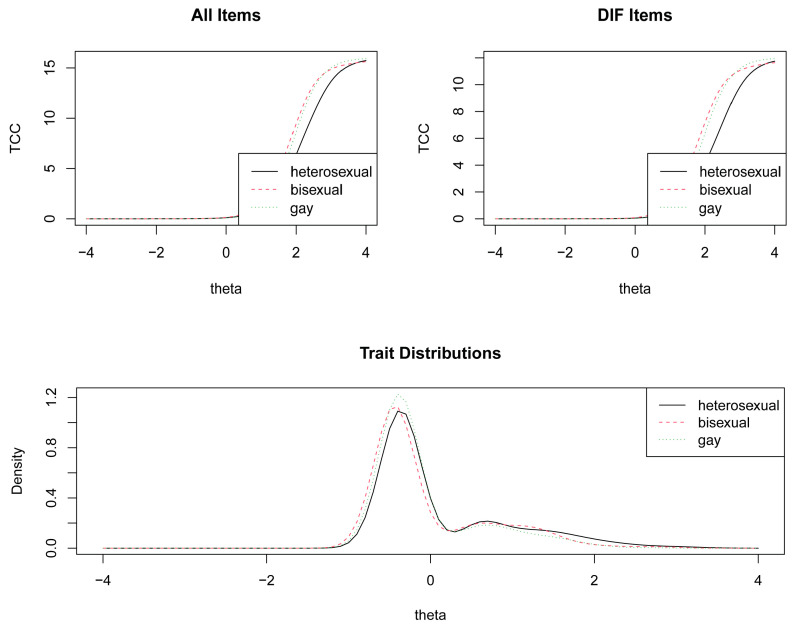
DIF of NATMI for sexual orientation.

**Table 1 ijerph-20-01820-t001:** Sociodemographic characteristics of the sample by sexual orientation.

	Heterosexual People	Bisexual People	Gay People
Gender *n* (%)			
Men	400 (50.00)	294 (42.40)	399 (49.90)
Women	400 (50.00)	400 (57.60)	400 (50.10)
Age *M* (*SD*)	40.74 (12.78)	29.35 (11.18)	34.91 (11.57)
Education level *n* (%)			
Primary Education	184 (23.00)	15 (2.20)	15 (1.80)
Secondary Education	121 (15.13)	113 (16.30)	134 (16.80)
University Degree	485 (60.62)	507 (73.10)	564 (70.60)
Other	10 (1.25)	50 (7.20)	79 (9.90)
Age of first sexual relationship *M* (*SD*)	17.87 (3.48)	16.79 (2.76)	17.81 (4.08)
Number of sexual partners *M* (*SD*)	11.44 (19.56)	15.84 (29.52)	29.87 (54.24)
Partner relationship *n* (%)			
Yes	585 (73.10)	393 (56.60)	480 (60.10)
No	215 (26.90)	301 (43.40)	319 (39.90)
Sexual activity in the relationship in the last 3 months *n* (%)			
Yes	550 (68.75)	226 (32.60)	326 (40.80)
No	35 (4.37)	11 (1.60)	35 (4.40)
Sexual activity without a partner in the last 3 months *n* (%)			
Yes	199 (24.87)	199 (28.7)	208 (26)
No	16 (2.00)	92 (13.30)	106 (13.30)
Current masturbation frequency *n* (%)			
More than once a day	39 (4.90)	42 (6.10)	41 (5.10)
Once a day	87 (10.90)	105 (15.10)	136 (17.00)
A few times a week	372 (46.50)	401 (57.80)	424 (53.10)
A few times a month	192 (24)	106 (15.30)	138 (17.30)
Less than once a month	75 (9.40)	35 (5.00)	50 (6.30)
Never	35 (4.40)	5 (0.70)	10 (1.30)

Note. *M* = mean, *SD* = standard deviation.

**Table 2 ijerph-20-01820-t002:** Descriptive analysis of items from SOS, HISF, and NATMI.

	Heterosexual People		Bisexual People		Gay People	
SOS (Range 1–7)	*M_e_*	*M* (*SD*)	*M_e_*	*M* (*SD*)	*M_e_*	*M* (*SD*)
Item 1	7	6.25 (1.33)	7	5.98 (1.47)	7	5.52 (1.96)
Item 2	7	6.53 (1.08)	7	6.59 (0.96)	7	6.56 (0.92)
Item 3	7	6.28 (1.25)	7	6.01 (1.38)	6	5.62 (1.79)
Item 4	7	6.11 (1.32)	7	6.14 (1.31)	7	6.04 (1.34)
Item 5	7	6.41 (1.24)	7	6.26 (1.39)	7	6.43 (1.22)
Item 6	7	5.89 (1.61)	7	6.05 (1.54)	7	6.09 (1.54)
HISF (Range 0–4)	*M_e_*	*M* (*SD*)	*M_e_*	*M* (*SD*)	*M_e_*	*M* (*SD*)
Item 1	4	3.42 (0.73)	3	3.27 (0.77)	4	3.42 (0.72)
Item 2	4	3.34 (0.85)	3	3.27 (0.86)	4	3.37 (0.83)
Item 3	3	2.80 (1.19)	3	2.84 (1.17)	3	2.77 (1.13)
Item 4	3	3.18 (1.00)	3	3.14 (1.00)	4	3.19 (0.97)
Item 5	3	2.87 (0.88)	3	2.67 (0.93)	3	2.84 (0.89)
Item 6	3	3.08 (1.11)	3	3.10 (1.03)	3	3.03 (1.11)
Item 7	3	2.72 (0.96)	3	2.66 (0.96)	3	2.70 (0.97)
Item 8	3	3.26 (0.80)	3	3.08 (0.88)	3	3.21 (0.81)
Item 9	3	2.91 (1.08)	3	2.87 (1.00)	3	2.89 (1.06)
Item 10	3	2.95 (1.10)	3	2.94 (1.09)	3	2.89 (1.06)
NATMI (Range 1–5)	*M_e_*	*M* (*SD*)	*M_e_*	*M* (*SD*)	*M_e_*	*M* (*SD*)
Item 1	1	1.09 (0.46)	1	1.06 (0.36)	1	1.04 (0.22)
Item 2	1	1.11 (0.41)	1	1.05 (0.27)	1	1.05 (0.27)
Item 3	1	1.07 (0.41)	1	1.04 (0.33)	1	1.02 (0.22)
Item 4	1	1.09 (0.42)	1	1.04 (0.26)	1	1.02 (0.17)
Item 5	1	1.19 (0.59)	1	1.30 (0.75)	1	1.16 (0.52)
Item 6	1	1.08 (0.38)	1	1.05 (0.25)	1	1.02 (0.14)
Item 7	1	1.22 (0.64)	1	1.12 (0.45)	1	1.07 (0.35)
Item 8	1	1.10 (0.45)	1	1.15 (0.49)	1	1.07 (0.35)
Item 9	1	1.17 (0.47)	1	1.12 (0.54)	1	1.17 (0.71)
Item 10	1	1.16 (0.55)	1	1.15 (0.54)	1	1.12 (0.49)

Note. *M_e_* = median. *M* = mean. *SD* = standard deviation. Sexual orientation: heterosexual (*n* = 800), bisexual (*n* = 694), and gay (*n* = 800).

**Table 3 ijerph-20-01820-t003:** CFA fit indices for SOS, HISF, and NATMI.

Model	*χ*^2^ (*df*)	CFI	TLI	RMSEA (90% CI)
SOS	148.806 (45) ***	0.948	0.913	0.053 (0.045, 0.061)
HISF	2451.967 (135) ***	0.936	0.918	0.099 (0.096, 0.103)
NATMI	342.429 (135) ***	0.950	0.935	0.025 (0.022, 0.028)

Note. SOS: Sexual Opinion Survey; HISF: Hurlbert Index of Sexual Fantasy; NATMI: Negative Attitudes Toward Masturbation Inventory. *** *p* < 0.001.

**Table 4 ijerph-20-01820-t004:** Measurement invariance across sexual orientation for SOS, HISF, and NATMI.

Invariance Model	*χ*^2^ (*df*)	CFI	TLI	RMSEA (90% CI)	ΔCFI	Invariance
SOS						
Configural	126.314 (45) ***	0.942	0.904	0.050 (0.041, 0.058)	-	Yes
Weak	119.137 (45) ***	0.936	0.922	0.045 (0.036, 0.054)	0.006	Yes
Strong	295.510 (45) ***	0.800	0.808	0.070 (0.062, 0.078)	0.136	No
Strict	500.485 (45) ***	0.631	0.719	0.085 (0.079, 0.082)	0.169	No
HISF						
Configural	2439.893 (135) ***	0.914	0.890	0.108 (0.104, 0.112)	-	Yes
Weak	1732.245 (135) ***	0.912	0.904	0.101 (0.097, 0.105)	0.002	Yes
Strong	1826.219 (135) ***	0.908	0.911	0.097 (0.093, 0.101)	0.004	Yes
Strict	1836.110 (135) ***	0.904	0.920	0.092 (0.088, 0.096)	0.004	Yes
NATMI						
Configural	342.286 (135) ***	0.948	0.933	0.023 (0.020, 0.026)	-	Yes
Weak	230.978 (135) ***	0.947	0.941	0.021 (0.017, 0.026)	0.001	Yes
Strong	413.200 (135) ***	0.856	0.862	0.033 (0.029, 0.036)	0.091	No
Strict	494.156 (135) ***	0.809	0.840	0.035 (0.032, 0.039)	0.047	No

Note. Sexual orientation: heterosexual (*n* = 800), bisexual (*n* = 694), and gay (*n* = 800). *** *p* < 0.001.

**Table 5 ijerph-20-01820-t005:** Differential item functioning: probabilities (*p*) associated with the models (M) of the regression and effect size.

	χ² (M2−M1)	*p*	χ² (M3−M2)	*p*	R^2^ Nagelkerke (M2−M1)	R^2^ Nagelkerke (M3−M2)
SOS						
Item 1	0.00	1.00	0.00	1.00	0.025	0.012
Item 2	0.33	0.847	0.91	0.634	0.0005	0.00
Item 3	0.00	1.00	0.00	1.00	0.031	0.009
Item 4	0.02	0.989	0.54	0.765	0.001	0.0002
Item 5	0.0004	0.999	0.01	0.993	0.006	0.003
Item 6	0.003	0.999	0.67	0.714	0.004	0.0003
HISF						
Item 1	0.0001	0.999	0.84	0.655	0.006	0.0001
Item 2	0.04	0.979	0.57	0.750	0.002	0.0003
Item 3	0.001	0.999	0.69	0.709	0.003	0.0002
Item 4	0.60	0.739	0.16	0.922	0.0003	0.001
Item 5	0.0004	0.999	0.58	0.750	0.005	0.0003
Item 6	0.004	0.997	0.25	0.885	0.002	0.0005
Item 7	0.78	0.678	0.93	0.628	0.0002	0.00
Item 8	0.002	0.999	0.47	0.789	0.003	0.0004
Item 9	0.64	0.726	0.30	0.859	0.0001	0.0004
Item 10	0.01	0.997	0.46	0.795	0.002	0.0003
NATMI						
Item 1	0.12	0.942	0.09	0.955	0.005	0.006
Item 2	0.21	0.902	0.78	0.677	0.003	0.0005
Item 3	0.41	0.815	0.04	0.979	0.003	0.012
Item 4	0.76	0.684	0.78	0.676	0.0008	0.0007
Item 5	0.00	1.00	0.00	1.00	0.043	0.009
Item 6	0.58	0.749	0.897	0.639	0.001	0.0003
Item 7	0.001	0.999	0.43	0.807	0.009	0.001
Item 8	0.00	1.00	0.02	0.991	0.060	0.005
Item 9	0.002	0.999	0.06	0.970	0.009	0.004
Item 10	0.00	1.00	0.02	0.989	0.015	0.004

## Data Availability

The data presented in this study are available on request from the corresponding author. The data are not publicly available due to privacy.

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
