# Peer review of "Measurement Invariance across Sexual Orientation for Measures of Sexual Attitudes"

_ijerph, 2023, doi:10.3390/ijerph20031820_

Round 1

Reviewer 1 Report

Line 98  It is not clear what "that conform sexuality work" is meant to say; is something missing?

Otherwise, nothing to note needing correction in this manuscript. 

It would be nice to break down age through number of sexual partners by gender in each of the three sexual orientation categories.

Reviewer 2 Report

To the editor: The present study aims to investigate the methodological qualities and validation of three scales that measure sexual behavior. The study is generally satisfactorily written, but I recommend that the authors revise some essential aspects before publication.

Page 2, Fifth paragraph. The authors mentioned, "In Judeo-Christian culture, masturbation has been traditionally stigmatized." Instead, a more general expression would be more appropriate because it is not only in the Asian population that such cultural perspectives on sexuality exist or exist.

The authors present the importance of sexual behaviors and their investigation, which is very good. However, the arguments why they chose precisely these three scales and why we should consider them the most relevant still need to be included. For example, considering the study's objective, the revised Sociosexual Orientation Inventory (SOI-R) could have been considered a relevant instrument (perhaps this aspect should also be introduced in future directions).

Also, the authors should insist on examining measurement invariance across sexual orientation of the Spanish versions of the three scales. It is necessary that these "measurement invariances" be presented in the introduction to clarify the usefulness of the paper.

Please add a short section with the theoretical and practical implications of the study.

Reviewer 3 Report

The manuscript presented the Measurement Invariance across Sexual Orientation for Measures of Sexual Attitudes.  The manuscript is sound in term of novelty. However, there are few suggestions that can improve the quality of manuscript.

1. Table 1., Table 4.  need to be in proper format and should be on a single page.

2. Paragraphs in the manuscript are too short.

3.  There are few grammatical errors which need to be address. 

4.  Results are appropriate in tabular form however graphical presentation of results will create more interest and significance of this research.

5. Too many keywords are given, keywords should be 3 to 5. 
